A reproducible approach to high-throughput biological data acquisition and integration

Börnigen Daniela 1 2 *
Moon Yo Sup 1
Rahnavard Gholamali 1 2
Waldron Levi 1 7
McIver Lauren 1
Shafquat Afrah 1
Franzosa Eric A. 1 2
Miropolsky Larissa 1
Sweeney Christopher 3
Morgan Xochitl C. 1 2
Garrett Wendy S. 2 4 5 6
Huttenhower Curtis 1 2 chuttenh@hsph.harvard.edu
1 Biostatistics Department, Harvard School of Public Health , Boston, MA , USA
2 The Broad Institute of MIT and Harvard , Cambridge, MA , USA
3 Dana-Farber Cancer Institute , Boston, MA , USA
4 Department of Immunology and Infectious Diseases, Harvard School of Public Health , Boston, MA , USA
5 Department of Medicine, Harvard Medical School , Boston, MA , USA
6 Department of Medical Oncology, Dana-Farber Cancer Institute , Boston, MA , USA
7 City University of New York School of Public Health, Hunter College , New York, NY , USA
Dessimoz Christophe
* Current affiliation: Department of Human Genetics, The University of Chicago, Chicago, IL, USA.

Electronic publication date: 2015 Mar 31
Publication date: 2015
Volume: 3
Electronic Location ID: e791
Received 2014 Oct 21; Accepted 2015 Feb 4
Copyright: © 2015 Boernigen et al.
Copyright year: 2015
Copyright holder: Boernigen et al.
License: This is an open access article distributed under the terms of the Creative Commons Attribution License, which permits unrestricted use, distribution, reproduction and adaptation in any medium and for any purpose provided that it is properly attributed. For attribution, the original author(s), title, publication source (PeerJ) and either DOI or URL of the article must be cited.
License URL: https://creativecommons.org/licenses/by/4.0/

Keywords: High-throughput data, Data integration, Data acquisition, Meta-analysis, Heterogeneous data, Reproducibility

Funding: National Science Foundation award CAREER DBI-1053486 National Science Foundation MCB-1453942 This work was supported by National Science Foundation award CAREER DBI-1053486, and a National Science Foundation grant (MCB-1453942) to Curtis Huttenhower. The funders had no role in study design, data collection and analysis, decision to publish, or preparation of the manuscript.

==============================
Modern biological research requires rapid, complex, and reproducible integration of multiple experimental results generated both internally and externally (e.g., from public repositories). Although large systematic meta-analyses are among the most effective approaches both for clinical biomarker discovery and for computational inference of biomolecular mechanisms, identifying, acquiring, and integrating relevant experimental results from multiple sources for a given study can be time-consuming and error-prone. To enable efficient and reproducible integration of diverse experimental results, we developed a novel approach for standardized acquisition and analysis of high-throughput and heterogeneous biological data. This allowed, first, novel biomolecular network reconstruction in human prostate cancer, which correctly recovered and extended the NFκB signaling pathway. Next, we investigated host-microbiome interactions. In less than an hour of analysis time, the system retrieved data and integrated six germ-free murine intestinal gene expression datasets to identify the genes most influenced by the gut microbiota, which comprised a set of immune-response and carbohydrate metabolism processes. Finally, we constructed integrated functional interaction networks to compare connectivity of peptide secretion pathways in the model organisms Escherichia coli, Bacillus subtilis, and Pseudomonas aeruginosa.

Introduction

The challenges of “big data” handling in modern biology have been highlighted repeatedly in recent publications (Lynch, 2008; Marx, 2013; Staff, 2011). The need for reproducible analyses of individual and integrated genome-scale data has resulted both in significant progress (Chan et al., 2008; Waldron, Coller & Huttenhower, 2012; Wirapati et al., 2008) and in substantial setbacks, such as the widely-publicized failed colon and ovarian cancer clinical trials (Dressman et al., 2007; Garman et al., 2008; Potti et al., 2006; Samuel Reich, 2011). Reproducibility in experimental biology is enforced by a combination of best practices and technical aids built up over decades of research: standardized protocols, reagents, recording practices, and equipment that ease the organizational burden necessary for high-quality research. Although parallel best practices have begun to be widely available for the equivalent processes in bioinformatic and data science investigations (Charaniya, Hu & Karypis, 2008; Noble, 2009; Wilson et al., 2014), they are not yet widely supported by standardized methods or computational environments enforcing reproducibility, which we develop here.

Reproducibly executing any single high-throughput biological study can be challenging, while reproducibly integrating multiple large-scale data sources presents an even greater hurdle. Meta-analysis of multiple genome-scale studies of the same biological system has proven to be a fruitful route for successful translational research, with over 500 such meta-analyses of human tumor gene expression alone now published (Chan et al., 2008; Wirapati et al., 2008). However, retrieving, standardizing, and curating such data is nontrivial, time-consuming, and error-prone. Integration of heterogeneous data can be an even more onerous process. For example, GEO (Barrett et al., 2011) and the BioGRID (Stark et al., 2011) each have over 2,000 citations as individual data sources, but fewer than 100 publications cite both (Warde-Farley et al., 2010). This lack of automatable systems for large-scale biological data handling and retrieval means that reliably incorporating such data into an analytical project is precluded without extensive computational expertise, and even in the best case, it is prone to error and challenging to reproduce (Gentleman, 2005; Ioannidis et al., 2009). This is particularly true when incorporating data from more than one source or type, or when linking data from public repositories to additional datasets from one or more individual labs. An example application, such as the identification of disease biomarkers, may involve establishing candidate markers by synthesis of published gene expression data, linking promising candidate markers to proteomic experiments, establishing putative biological mechanisms through databases of regulatory and protein interactions, and finally validating only the best candidates manually in the lab (Lu et al., 2005; Rifai, Gillette & Carr, 2006; Van ’t Veer et al., 2002). The advantages of such holistic approaches are obvious, but greatly expand the complexity of analysis.

Data retrieval systems have recently been developed that facilitate this process for consistent, large-scale access to individual ’omics repositories. The Gene Expression Omnibus (GEO) (Barrett et al., 2011; Barrett et al., 2013) and ArrayExpress (Rustici et al., 2013), for example, represent two of the largest functional genomics resources for transcriptional data, and both feature elegant web-based query systems (e.g., the Gene Expression Atlas (Kapushesky et al., 2012)) and programmatic access (e.g., GEOQuery (Davis & Meltzer, 2007)). Similarly, molecular interaction databases such as IntAct (Kerrien et al., 2012; Orchard et al., 2014) and the Biological General Repository for Interaction Datasets (BioGRID) (Chatr-Aryamontri et al., 2013; Stark et al., 2011) typically provide access to selected subsets of protein–protein interaction networks by way of taxonomic, technological, or temporal metadata. Each of these interfaces is database-specific, although they provide their data downloadable in the PSI-MI standard format; in general, expression data retrieved programmatically in R must be manually converted and combined with interaction data. Although some systems such as DAS (Dowell et al., 2001) provide a uniform interface for multiple ’omic data types, accessing data from most public repositories presents a major challenge in the compatibility of the resulting data formats, mapping between different gene identifier conventions, and the curation of inconsistently-annotated experiments. Manual parsing and conversion of experimental attributes and values is tedious and error-prone (Baggerly & Coombes, 2009), and the large number of normalization and preprocessing methods in use make comparison of different existing studies difficult or even impossible (Sims, 2009). This suggests the need for a unified approach capable of automatically retrieving and reproducibly processing data from multiple different ’omics repositories.

Previous related efforts for reproducible research include (1) general workflow systems for broad bioinformatic analysis that can be tuned for data retrieval, (2) tools enabling programmatic access to public databases, and (3) curated alternatives to the largest public experimental databases. General workflow systems, such as Kepler (Barseghian et al., 2010) and Taverna (Hull et al., 2006), facilitate the construction of reproducible programmatic workflows. In many cases they include programmatic access to biological databases, but subsequent data and metadata format harmonization must be implemented separately by the user. Tools that allow programmatic access to public databases, such the specialized interfaces provided by GEOquery (Davis & Meltzer, 2007) and Ensembl BioMarts (Kinsella et al., 2011), are among the most common. GEOquery is specific to automated queries and downloads from one particular database, and while queries can be written across multiple BioMart databases, even the latter tool includes no facilities for normalization, integration, or meta-analysis of the resulting independent query responses. Finally, curated alternative data management and analysis platforms, such as EZArray (Zhu & Xu, 2008), BxArrays™, and ONCOMINE (Rhodes et al., 2004), provide web interfaces in which a user may perform only pre-selected analyses of the provided gene expression datasets. Another open data management and access platform, inSilicoDB (Taminau et al., 2011), allows programmatic download of gene expression data curated by maintainers or contributors from the community, although users may only retrieve one experiment at a time. None of these tools facilitate bulk download of ’omics data from multiple sources for efficient processing on local computing resources, and most enable only per-repository data acquisition. A logical next step for large scale, multi’omic data acquisition for meta-analysis and heterogeneous data integration is again a batch-driven data acquisition system spanning multiple sources.

Motivated by this need to access and integrate diverse sources of experimental results, we developed an approach for reproducible processing of high-throughput and heterogeneous data. The system, implemented as the ARepA (Automated Repository Acquisition) software environment, enabled us to rapidly and systematically retrieve, integrate, and analyze large ‘omics datasets from multiple sources in a consistent and robust fashion. Here, we first undertook a study of the NFκB signaling pathway in the context of human prostate cancer by integrating ten tissue-specific human expression datasets across six platforms. This not only recovered but extended the currently known molecular network. Next, due to burgeoning recent interest in microbiome studies and the considerable knowledge gaps regarding host processes are dependent upon the gut microbiota, we investigated transcripts that were differentially expressed in the germ-free mouse colon, identifying and integrating six gene expression datasets. PPAR-alpha, RXR-alpha, and several other immune- and carbohydrate metabolism-related genes were detected as differentially expressed in this gene expression meta-analysis. Finally, we integrated protein–protein interactions, gene expression, and regulatory networks for three very diverse model bacteria (Bacillus subtilis, which is Gram-positive; Escherichia coli, which is Gram-negative, and Pseudomonas aeruginosa, which is Gram-negative and pathogenic), correctly identifying differentially-conserved peptide secretion system pathways.

Each of these three studies required rapid, reproducible, and diverse data collection and integration, and thus demonstrates the utility of ARepA for any lab wishing to combine its own data with public data repositories. We provide an implementation of ARepA that can access ’omics data from multiple data sources in a consistent and robust fashion. Furthermore, it can be configured to process data only from certain sources, of specified types, or from particular organisms of interest, depending on the user’s interest and requirements. Currently, it supports seven repositories: the Gene Expression Omnibus (GEO) (Barrett et al., 2011; Barrett et al., 2013), IntAct (Kerrien et al., 2012; Orchard et al., 2014), BioGRID (Stark et al., 2011), RegulonDB (Gama-Castro et al., 2011), STRING (Szklarczyk et al., 2011), Bacteriome (Su et al., 2008), and MPIDB (Goll et al., 2008) (Table 1), from which data are retrieved, processed and standardized for any organism of interest. At present, this produces more than 2 M interactions from ∼600 K physical interactions (Bacteriome, BioGRID, Intact, and MPIDB), ∼4 K regulatory interactions (RegulonDB) and 1.5 M functional associations (STRING), as well as more than 2.7 K expression data sets covering more than 800 K samples (GEO). When retrieving data from six model organism, ARepA currently produces >630 K interactions and >15 K gene expression data sets (>530 K samples), accompanied by corresponding metadata and derived molecular networks (Table 2 and Fig. 1).

Figure 1 ARepA is an extensible, modular Automated Repository Acquisition system for reproducible biological data acquisition and processing.

ARepA is a framework for reproducible biological data mining and analysis. It can retrieve heterogeneous data from multiple public repositories in a uniform environment and format, currently allowing configurable data access for any organism(s) to the Gene Expression Omnibus (GEO) (Barrett et al., 2011), IntAct (Kerrien et al., 2012), BioGRID (Stark et al., 2011), RegulonDB (Gama-Castro et al., 2011), STRING (Szklarczyk et al., 2011), Bacteriome (Su et al., 2008), and MPIDB (Goll et al., 2008) databases. Using ARepA includes the following steps: (i) user input, (ii) data processing, and (iii) output formatting. The input phase is the only aspect of the ARepA pipeline that requires direct user oversight. The user’s input to ARepA can be as simple a list of organisms-of-interest; ARepA then uses this list as a query for recovering interactome network and gene expression data specific to those organisms. Advanced users also have the option of providing custom gene mapping files, metadata, and/or normalization schemes, as well as fine-tuning the list of data sources to be searched. The data processing phase is divided into a series of automated steps in which raw interactome network and gene expression data are downloaded, converted to a common gene-naming scheme, and normalized for between-dataset comparison. During this phase, integrated gene expression data are analyzed for co-expression relationships, which contributes an additional co-expression network to the final network output. All network data are provided for the user as text files, while all expression data and associated metadata are saved as individual text files and as an R data file. The bottom panel illustrates how generated data can be integrated by subsequent network (see prostate cancer and bacterial studies) or expression (see murine differential gene expression) meta-analysis. For example, network integration is a convenient way to combines multiple datasets of different types and sources, such as co-expression, physical gene interactions, regulatory interactions, or posttranslational modification, into one functional network.

Table 1 ARepA extracts heterogeneous data from several distinct data types: physical interactions (Bacteriome, BioGRID, MPIDB, and IntAct), functional associations (STRING), gene expression (GEO), and regulatory interactions (RegulonDB).

All repositories can be configured to retrieve only a specified range of taxonomic identifiers, and individual dataset IDs can be included or excluded from applicable repositories. Output data includes standardized data files in multiple formats (text tables, interaction networks, and/or R packages) as well as metadata for all data types and datasets.

Database	Interaction data type	# of species (as of 03/2013)	# of interactions (as of 03/2013)	Reference	Metadata output type	
Bacteriome	Physical	1	3,888	(Su et al., 2008)	Taxonomy ID	
BioGRID	Physical	32	349,696	(Stark et al., 2011)	Platform, experiment type, taxonomy ID	
IntAct	Physical	278	239,940	(Kerrien et al., 2012; Orchard et al., 2014)	Platform, experiment type, taxonomy ID	
MPIDB	Physical	250	24,295	(Goll et al., 2008)	Platform, experiment type, taxonomy ID	
RegulonDB	Regulatory	1	4,096	(Gama-Castro et al., 2011)	Taxonomy ID	
STRING	Functional	1,133	1,640,707	(Szklarczyk et al., 2011)	Taxonomy ID, mode	
GEO	Expression	1, 967		(Barrett et al., 2011; Barrett et al., 2013)	Taxonomy ID, experiment type, pubmed ID, platform, title, glossary, channels, conditions	

Table 2 ARepA currently yields more than 2.7 K expression data sets from GEO and more than 2 M interactions.

(A) ARepA currently yields more than 2.7 K expression data sets from GEO covering more than 800 K samples. (B) ARepA currently yields more than 2 M interactions: 600 K physical interactions (Bacteriome, BioGRID, IntAct, and MPIDB), 4 K regulatory interactions (RegulonDB), and 1.5 M functional associations (STRING).

(A)	
Datatype	Human	Mouse	Yeast	E. coli	B. subtilis	P. aeruginosa	
Number of datasets and samples (as of 03/2013)	
Expression datasets	10,575	3,940	162	184	15	73	
Expression samples/conditions	467,369	57,813	2,374	1,955	199	802	
(B)	
Datatype	Database	Human	Mouse	Yeast	E. coli	B. subtilis	P. aeruginosa	
Number of interactions (as of 03/2013)	
Physical	Bacteriome	na	na	na	6,741	na	na	
	BioGRID	49,794	1,815	103,475	0	0	0	
	IntAct	46,961	1,025	30,205	10,992	407	2	
	MPIDB	na	na	na	1,449	118	20	
Regulatory	RegulonDB	na	na	na	71	na	na	
Functional	STRING	226,986	1,129	22,222	85,608	2,931	44,032	

Material and Methods

Case studies

Human prostate cancer data retrieval and molecular network integration

We screened ARepA’s GEO metadata database for human prostate cancer and prostate tissue conditions matching “prostate cancer” in human gene expression profile studies with at least 6 samples each, identifying ten relevant datasets from six different platforms (Affymetrix, Agilent, and CNIO, Table S1). Identifying, processing, and standardizing these datasets in ARepA, accompanied by computing co-expression networks, required <1 h running time in total (on a 2.9 GHz Intel Core i7 16 GB machine). These data were then meta-analyzed to a single prostate cancer specific functional network using unsupervised data integration averaging across normalized co-expression values (z-scores) (Huttenhower et al., 2009) (Text S1). Next, we used this integrated network to predict genes highly functionally related to NFκB gene family (NFκB1, NFκB2, RELA, RELB, REL) in prostate cancer by extracting a high-confidence subgraph based on the probabilistic graph search algorithm as described in Huttenhower et al. (2009); Myers et al. (2005). This starts from a user-defined set of query genes and identifies k additional neighbours in the network that are connected with high confidence to the original query genes (Huttenhower et al., 2008). Here, we defined a query gene set of twelve genes from the NFκB signaling pathway in BioCarta (Table S4) and a neighbourhood size of k = 10, resulting in a high-confidence NFκB signaling network containing 22 genes in total as illustrated in Fig. 2.

Figure 2 MEN1 and ACBD6 associated with the NFκB signaling pathway in human prostate cancer.

High confidence subgraph extracted from a functional network integrating ten prostate cancer specific gene expression data sets from GEO (Table S1). This subnetwork was generated using a seed gene set of ten genes from the NFκB signaling pathway in BioCarta (blue circles). Nine genes (black circles) were immediately recovered that are also known to be involved in NfκB signaling. Additional genes represent candidates implicated in NFκB involvement during prostate cancer, in particular MEN1 and ACBD6.

Murine expression data retrieval and meta-analysis

We conducted a differential expression meta-analysis of genes and pathways up- or down-regulated in the germ-free (without microbes) murine gut (Fig. S1 and Text S1). We used six case-control datasets containing intestinal tissue and germ-free versus wild type comparisons, identified through ARepA’s metadata screen by matching “germ-free,” “wild type,” and “intestinal tract” in mouse gene expression profiles with at least four samples each (Table S2). This again required <1 h total running time (on a 2.9 GHz Intel Core i7 16 GB machine). To integrate the resulting ARepA R data files, we first computed log fold changes and confidence intervals (2.5% and 97.5%) for all genes between germ-free gut and wild type gut within each dataset using the R/limma package (Smith, 2005), resulting in ∼3,600 differentially expressed genes. Next, we performed a meta-analysis using the R/metafor package (Viechtbauer, 2010) on the six datasets by applying a random-effects model on the fold changes with default options, fitting the model with the restricted maximum-likelihood estimator (REML). The false discovery rate was controlled by the Benjamini–Hochberg method (Benjamini & Hochberg, 1995). We finally tested all genes and their resulting meta-p-values for gene set enrichment (Subramanian et al., 2005) in KEGG and BioCarta pathways. This resulted in two pathways that were significantly enriched for upregulated genes under germ-free conditions (using 1,000 permutations, Fig. 3), while 15 pathways were enriched for downregulated genes (Table S3).

Figure 3 Differential expression meta-analysis of germ-free versus conventional mice.

ARepA metadata allowed the identification of six murine gene expression datasets with intestinal tissue from paired germ-free and conventional mice (Table S1). The automatically generated R expression sets were meta-analyzed using R/limma (Smith, 2005) and R/metafor (Viechtbauer, 2010) through a random-effects model, revealing the Ppar-α signaling pathway as one of several differentially regulated gene sets. In (A) the fold changes are presented for all significantly differentially expressed genes from this pathway in individual datasets, and (B/C) show the corresponding forest plots for the Ppar-α and Rxr-α genes, which are consistently upregulated in these datasets.

Data integration for bacterial model organisms

We retrieved all gene expression data (GEO) and gene interaction data (IntAct, MPIDB, RegulonDB, Bacteriome and STRING, Table 2A) from ARepA for Bacillus subtilis, Escherichia coli and Pseudomonas aeruginosa. These were processed as above and standardized onto a shared gene identifier (UniRef90). Within each bacterium, these data were integrated into a species-specific functional network using an unsupervised data integration averaging across normalized co-expression values (z-scores) (Huttenhower et al., 2009) and extracting a high-confidence subgraph based on the probabilistic graph search algorithm (Huttenhower et al., 2009; Myers et al., 2005) (Text S1). This query (Huttenhower et al., 2008) started from a user-defined set of genes of the sec and tat genes and genes from the Type I, Type II, Type III, Type V, and Type VI secretion systems (Table S4) and used a neighbourhood size of k = 5 for each subgraph extraction. As Gram-positive bacteria use only the general secretory (sec) pathway and the tat pathway, we used two tat genes and nine sec genes as the query gene set in B. subtilis. In contrast, E. coli, a Gram-negative bacterium,  employs an additional secretion system (Type II secretion system) and has one type I gene (TolC); thus, we used four tat genes, twelve sec genes, one type I gene, and twelve type II genes as a query gene set in E. coli. P. aeruginosa, another Gram-negative bacterium, additionally utilizes the type I, type III , type V, and type VI secretion systems, so we defined our query gene set of three tat genes, nine sec genes, two type I genes, eleven type II genes, twelve type III genes, one type V gene, and eight type VI genes for retrieving a species-specific high-confidence network in P. aeruginosa. We recovered networks of 16 genes for B. subtilis (Fig. 4A), 34 genes for E. coli (Fig. 4B), and 49 genes for P. aeruginosa (Fig. 4C), which showed clustering of secretion types conserved across species.

Figure 4 Integrated molecular networks for comparative microbial functional genomics.

ARepA allowed the retrieval of standardized gene expression and interaction data for three microbial species based on a shared gene identifier to assess functional differences in conserved and non-conserved secretion pathways. High-confidence subgraphs were extracted from species-specific integrated functional networks around genes from species-specific secretion pathways to identify highly functionally related gene clusters within each individual system. These subgraphs represent gene clusters of Sec and Tat genes in B. subtilis (A), sec, tat, and Type II genes in E. coli (B), and sec, tat, Type II, Type III, and Type VI genes in P. Aeruginosa (C). From each of these species-specific molecular networks we recovered highly functionally related gene clusters and conserved and non-conserved components from the peptide secretion system.

Implementation

ARepA is designed around “dedicated” and “shared” processing modules. Dedicated modules download and process raw data from individual public repositories, while shared modules process the resulting data into standardized formats and aid key steps common to multiple dedicated modules. This enables ARepA to customize the unique aspects of individual public repositories’ interfaces while sharing consistent information across all repositories, such as systematic identifiers for taxonomy, genes, and metadata.

Each repository in ARepA is implemented as one dedicated module, all of which share a similar design template. The main shared module typically executed in any dedicated module is the GeneMapper, which installs and initializes the BridgeDB package (BatchMapper command line tool) (Van Iersel et al., 2010) to map gene identifier consistently across all data types after data retrieval. Later in each repository’s processing steps, the shared Package module is executed to export all output data (such as expression datasets, networks, and metadata) into R data files, in addition to tab-delimited text, enabling an easy access to the data for downstream analysis.

ARepA is implemented in Python and is based on the SCons tool for dependency-driven modular computing. SCons ensures that data processing is efficient (i.e., on an as-needed basis), modular (necessary processing steps for each repository and data product are isolated), and simple (a single command is used to retrieve and integrate all requested data). Therefore, new data are only generated when necessary due to changes in an upstream element, such as new data versions or availability. These changes may include updates to remote data or metadata (e.g. file size or timestamp changes), modifications to the requested processing steps (scripts or configuration files), or changes to the ARepA configuration itself.

ARepA filesystem and configuration

ARepA’s implementation consists of a filesystem hierarchy in which the top level coordinates overall execution and contains shared code and documentation, and each subdirectory contains the custom processes necessary to access a specific data repository. Currently, ARepA houses seven sub-modules for its default repositories: GEO (Barrett et al., 2011), IntAct (Kerrien et al., 2012), BioGRID (Stark et al., 2011), Bacteriome (Su et al., 2008), RegulonDB (Gama-Castro et al., 2011), MPIDB (Goll et al., 2008), and STRING (Szklarczyk et al., 2011). Each module in ARepA contains two groups of persistent elements: (1) driver scripts (SCons-based SConstruct and SConscript files) and (2) module-specific directories. Each driver script launches all processes required to subsequently launch child processes in submodules. Module-specific directories are organized consistently to include: (1) src folders with module-specific source for data processing, (2) etc folders contain all configurable and user-defined information of the module, (3) tmp folders store intermediate downloaded files before processing and standardization, (4) doc folders contain documentation specific to module, and (5) data folders store the output of the final processed and standardized data.

ARepA includes a global configuration file that optionally limits the processing of all data to a subset of organisms/clades from the NCBI taxonomy (Sayers et al., 2009). Additionally, module-specific configuration files may exist to provide more specific configuration settings (see below). Notably, the GEO module has configuration files to change the type of datasets retrieved (GSE/GDS/GPL/GSM), how microarray values are normalized, which standard gene identifiers are used, the addition of manually curated metadata for specific datasets, and regular expression searches of available data. Due to ARepA’s modular design, each repository’s data can be handled independently by its corresponding driver script without the need of processing all available repositories, while also incorporating global configuration information from higher levels in the hierarchy.

Configuration files and formats

Procedural configuration files allow different combinations of normalization or processing to be carried out on different ARepA data subsets. For example, the default configuration file for a pre-processing step in GEO contains the string “affy::rma”, which instructs ARepA to normalize expression data with the Robust Multi-array Average (RMA) method (Irizarry et al., 2003). Another option is the MAS5 method (Affymetrix, 2002), which can be called with the “affy::mas5” string; complete documentation is provided in the ARepA online manual. Override configuration files, conversely, provide additional metadata (typically manually curated) to be associated with downloaded data. For instance, providing a manually curated override metadata file for the GEO dataset GSE10183 in the corresponding configuration directory (GEO/etc/manual_curation/GSE10183.txt) will instruct ARepA to bypass the default metadata pipeline and use the override configuration file as a gold standard instead.

Top-level procedural configuration allows the user to configure the taxonomy and gene identifier format of interest. Repository-specific procedural configuration includes the specification of preprocessing functions, regular expression searches for gene mapping, and including/excluding datasets. Repository-specific override configuration allows the incorporation of user-curated metadata and gene identifier maps along with automatically generated metadata.

Data and metadata

ARepA provides expression, gene interaction data, and metadata as output. ARepA handles three expression data file types: raw, mapped, and normalized. Raw expression data are the default as acquired from the repository, while mapped data include probe IDs assigned to gene names. Finally, normalized expression data are the output of a process consisting of a z-score transformation and missing value imputation as implemented by Sleipnir (Huttenhower et al., 2008). Here, raw values are normalized by transforming the vector of expression values for each gene to z-scores (i.e., mean zero and standard deviation one). Subsequently, missing values are imputed with a weighted average of the corresponding values in their 10 nearest neighbors (weighted by relative Euclidean distance). To aid in assessing data quality, ARepA generates a standardised Quality Control (QC) report using the Bioconductor package arrayQualityMetrics (Kauffmann, Gentleman & Huber, 2009).

ARepA’s reliance on the SCons build system for dependency-driven processing ensures that remote data and metadata are only re-downloaded when changed. There are two corresponding types of interaction data—raw and mapped—which contain default gene pairs and mapped gene pairs, respectively. Currently, gene expression data is retrieved uniquely by GEO, while interaction data is available for all repositories. Corresponding metadata and a co-expression network (Huttenhower et al., 2008) are also generated which are summarized in a portable R data file. ARepA generates interaction and regulatory networks with corresponding metadata information for all internal modules.

ARepA’s metadata model generalizes to accommodate essentially any source repository information and comprises an unordered collection of nested key-value pairs. Each final data file is associated with exactly one metadata file, which includes all information that is available about a dataset through curation or each repository’s automated retrieval. Standardized metadata across repositories include each dataset’s taxonomic identification, publication identifier, platform identifier, experimental information (see Table 1 and Text S2) and provenance pertaining to ARepA’s internal processes, such as a log of each command run, its version, MD5 checksums, and the success status of the gene standardization for a specific dataset. If requested, curated metadata can be combined with generated metadata (e.g., when a table of per-sample detailed clinical covariates is available).

Reproducible data processing

Although ARepA will operate using default configuration parameters out-of-the-box, it provides four main input interfaces through which its behavior can be modified: global taxonomic targeting, global gene identifier standardization, per-repository dataset ID restriction, and curated metadata. To ensure standardized data retrieval, ARepA contains global configuration files which regulate the kind of data that will be produced in each repository. Either all data from every repository can be retrieved, or the species for which data is requested can be limited by NCBI Taxonomy identifier or taxon name. Likewise, any target gene identifier formatted supported by GeneMapper can be specified to ensure standardized output data across all repositories.

Each repository has additional custom configuration parameters that specify which datasets are to be included or excluded from processing. These are specified using the unique dataset identifiers for each repository, e.g. GSD or GSE dataset IDs for GEO. Finally, metadata available for each dataset from any repository is included automatically, but all repositories also allow curated metadata to be automatically merged with these provided metadata. Curated metadata are automatically detected if provided in a standardized location within each repository module and, like repository-provided metadata, can include key-value pairs in pre-specified templated formats (e.g. numerical keys, enumerated values, free text, and so forth). This allows for the construction of truly customized high-throughput meta-analytic studies.

ARepA’s main outputs are one or more data file(s) (text format) and a metadata file (Python pickle format) per dataset (Text S2); this is Python’s native serialization format and is efficient to use and query for rapid database joins and searches. The output data consists of (1) a tab-delimited text matrix when appropriate (e.g., gene expression) of the raw data as downloaded from the repository, the processed data, and the standardized data after gene identifier mapping; (2) a tab-delimited network edge list of physical interactions, regulatory interactions, or co-expression from normalized and standardized data; and (3) an R data file integrating all processed and standardized output and metadata files for each expression data set as a Bioconductor ExpressionSet in one R file, containing experimental data (experimentData: “Experimenter name,” “Laboratory,” “Contact information,” “Title,” “PMIDs,” “Abstract,” and “notes”), annotation (annotation: platform GPL identifier), phenotypic data (phenoData: per-condition metadata (GSE only), as obtained from GEO by default or the custom curated per-sample metadata if provided at runtime), and gene expression data (exprs: expression data with mapped (unmapped for GDS) probeset identifiers). All raw, processed and standardized output data and metadata files are stored in one directory per dataset within the corresponding repository’s data directory, and custom processing modules can be included in any repository to produce additional outputs (or modify the default outputs). Each repository’s data output directory consists of a structure of subdirectories, organizing the output data into dataset-, taxa- and publication-specific subfolders that contain their corresponding output files (as described above) for an easy and intuitive access to specific output data of interest.

Adding repository modules

ARepA’s modular design allows the seamless integration of custom submodules provided by the user. A user can extend ARepA to include new repositories by creating a subdirectory, supplying a driver script and one or more execution scripts, specifying steps to obtain the raw data and, if needed, passing the resulting files downstream for additional processing. These process the raw file types provided from the repository into desired standardized formats and can access shared ARepA resources to perform common tasks such as gene ID standardization.

A typical submodule workflow includes: (1) Download the raw file of a dataset from a repository and store in the tmp directory; (2) Parse procedural configuration information specified in the etc folder and determine computational steps necessary to produce the output; (3) Execute relevant scripts in the src in the combination and order as specified by procedural configuration; (4) If override configuration exists, replace default files with user-provided files. Programmatic wrappers to these system calls are available internally for facilitation of use. For a thorough walkthrough of this feature, consult the documentation provided with the system.

Results

Motivated by the need to reproducibly access and unify diverse data resources, we developed a novel approach for the retrieval, normalization, and meta-analysis of high-throughput biological data. In this study, we applied our system, ARepA (Automated Repository Acquisition), to three integrative “big data” analyses: an integration of gene networks in human prostate cancer to identify interactors with the NfκB pathway, a differential expression meta-analysis of genes and pathways up- or down-regulated in the germ-free murine colon, and an integrative analysis of heterogeneous data from three model microbes to identify differentially conserved peptide secretion system pathways.

ARepA is implemented as a publicly available software package, currently providing reproducible and automated data and metadata processing for the Gene Expression Omnibus (GEO) (Barrett et al., 2011), IntAct (Kerrien et al., 2012), BioGRID (Stark et al., 2011), RegulonDB (Gama-Castro et al., 2011), STRING (Szklarczyk et al., 2011), Bacteriome (Su et al., 2008), and MPIDB (Goll et al., 2008). Investigators can retrieve normalized data and metadata in standardized formats from these repositories for any organism of interest. For six model organisms studied here in detail (human, mouse, yeast, E. coli, B. subtilis, and P. aeruginosa), it currently yields data comprising more than 630 K interactions: 250 K physical interactions (Bacteriome, BioGRID, IntAct, and MPIDB), and 380 K functional associations (STRING), in addition to approximately 15 K expression data sets covering more than 530 K samples (GEO) with standardized formats and gene identifiers. Metadata and network data are also automatically produced for each dataset and can be summarized in a portable R data file (Fig. 1).

MEN1 and ACBD6 are associated with the NFκB signaling pathway in human prostate cancer

The NFκB gene family of transcription factors plays an important role in immune response (Hayden, West & Ghosh, 2006) and can promote the growth and survival of solid and haematological malignancies, including prostate cancer (Davis, Kucuk & Sarkar, 1999; Perkins, 2012). In order to investigate this key gene family in human prostate cancer, we used ten prostate cancer gene expression datasets (Table S1) to construct a single prostate cancer-specific functional network (see Methods); the entire process of querying, downloading, normalizing, standardizing and integrating these datasets took <1 h total running time (on a 2.9 GHz Intel Core i7 16 GB machine). To identify genes highly related to NFκB gene family in human prostate cancer, we extracted a high-confidence subgraph from this integrated network around twelve NFκB signaling genes in BioCarta (see Methods, Table S4). This query process retrieved seven additional genes known from prior work to be involved in NFκB signaling (Davis, Kucuk & Sarkar, 1999; Perkins, 2012) (Fig. 2): Tumor protein p53 (TP53), TNF receptor-associated factor 2 (TRAF2), receptor (TNFRSF)-interacting serine-threonine kinase 1 (RIPK1), mitogen-activated protein kinases (MAP3K14 and MAP2K1), and interleukins (IL1B and IL10).

Two newly-associated genes with potential relationships to NFκB and prostate cancer were also recovered: Menin (MEN1) and acyl-CoA binding domain containing 6 (ACBD6). MEN1 is a putative tumor suppressor known to physically interact with NFκB proteins (Heppner et al., 2001). Here, we observed that in prostate cancer, MEN1 was significantly co-expressed with TP53 and IκB kinase (IKK), specifically the gamma subunit (Fig. 2). The IκB complex phosphorylates NFκB to activate the cellular response to inflammation (Bouwmeester et al., 2004; Hacker & Karin, 2006; Wen et al., 2000), suggesting that MEN1 may represent a novel upstream regulator of the pathway in prostate cancer.

ACBD6 is a poorly-characterized protein that has been implicated in few specific areas; one of these is growth of Chlamydia trachomatis, for which it is essential in cell culture (Soupene et al., 2012). C. trachomatis has been implicated as a microbial driver in prostate carcinogenesis and has also been shown to elicit chemokine production from immune cells and non-immune cells isolated from the prostate (Oberti et al., 2011). Our prediction suggests a more specific role for ACBD6 in prostate cancer as a potential suppressor of NFκB activity and downstream inflammation, as it was highly co-expressed with the two NFκB inhibitors, NFκB IA and NFκBIE (Pringle et al., 2012; Wu et al., 2006) (Fig. 2). This hypothesis is consistent with ACBD6’s previously-reported role in bacterial growth if it represents one mechanism by which the antimicrobial response is repressed through suppression of NFκB.

Ppar-α and Rxr-α are consistently differentially expressed in a germ-free mouse intestinal meta-analysis

We next used ARepA to conduct a differential expression meta-analysis of genes up- or down-regulated in the germ-free murine gut (Fig. S1). ARepA metadata from GEO included six case-control gene expression datasets containing intestinal tissue and comparing germ-free and wild type mice (Table S1). The process of identifying these datasets from throughout GEO, obtaining them, and normalizing their values, gene identifiers, and metadata took <1 h total running time (on a 2.9 GHz Intel Core i7 16 GB machine). ARepA made the retrieved datasets available as R data files, which were then meta-analyzed (see Methods). This identified ∼3,600 genes differentially expressed in the germ-free gut (Table S5); these were then tested for gene set enrichment in KEGG and BioCarta pathways (Subramanian et al., 2005). Under germ-free conditions, two pathways were significantly enriched for upregulated genes, and 15 pathways were enriched for downregulated genes (Table S3).

Almost all downregulated pathways under germ-free conditions were immune-related, including cytotoxic T lymphocyte-mediated immune response, T cytotoxic and T helper cell surface molecules, and B cell and T cell receptor signaling (Table S3). Germ-free mice physiologically exhibit profound immune system deficiencies, including smaller lymph nodes, Peyer’s patches, reduced immunoglobin production (Rahija, 2007), reduced numbers of CD4+ T cells (Mazmanian et al., 2005) (particularly in the intestinal lamina propria (Round & Mazmanian, 2009)) and reduced numbers of intraepithelial CD8+ T cells (Round & Mazmanian, 2009). This severe dysregulation of immunity under germ-free conditions is thus consistent with existing literature and correctly detected by ARepA’s automated meta-analysis.

The two upregulated pathways involved peroxisome proliferator-activated receptors (Ppar), which are nuclear hormone receptors that are activated by fatty acids and their derivatives. For example, in germ-free mice, the Ppar signaling pathway from KEGG (mmu03320) was enriched for 11 significant differentially expressed genes across all datasets (Fig. 3A), eight of which were Ppar-α target genes in lipid metabolism (Rakhshandehroo et al., 2010). Ppar-α is a major transcriptional regulator of lipid metabolism and is expressed in the liver and the intestinal tract. Although levels of bacterially-generated short-chain fatty acids are markedly decreased in the germ-free mouse intestine, Ppar-α is upregulated (Aoyama et al., 1998; Bookout et al., 2006; Hoverstad, Midtvedt & Bohmer, 1985; Hussain, 2013; Mukherji et al., 2013), as confirmed in our meta-analysis (Fig. 3B). The upregulation of Ppar-α promotes the uptake, utilization, and catabolism of fatty acids by upregulating genes involved in lipid metabolism (Aoyama et al., 1998). Accordingly, downstream target genes, such as Acsl1 and Acsl5 (fatty acid binding and activation), Apoa1 (fatty acid uptake and metabolism), Acadm (fatty acid degradation), Fads2 (lipogenesis), or Nr1h3 (cholesterol transport and metabolism) were all consistently upregulated in expression in our meta-analysis (Fig. 3A), suggesting their downstream activation in lipid metabolism not only in liver and skeletal muscle (Dreyer et al., 1992; Rakhshandehroo et al., 2010), but also in the intestinal tract. In addition, our meta-analysis revealed consistent upregulation of Rxr-α (Fig. 3C), a nuclear receptor that heterodimerizes with Ppar-α (Rakhshandehroo et al., 2010) and mediates the biological effects of retinoids, which play an important role in immune homeostasis in the intestine. As Ppar-α and Rxr-α synergistically regulate the target genes of Ppar-α within the Ppar signaling pathway from KEGG (mmu03320), we suggest that this synergistic upregulation of both genes in the intestine drives the downstream expression of fatty acid transport, lipid binding, lipid activation, and lipid degradation in the intestinal tract. This finding suggests lipid metabolism in addition to the immune system as a highly dysregulated systems in germ-free mice, potentially reflecting the altered metabolic challenges/relative energy deprivation the germ-free mouse confronts, as mice rely on their gut microbiota for optimal energy extraction from the diet.

Conserved and non-conserved secretion pathways in evolutionarily diverged bacteria

Although their genomes are relatively simple, bacteria represent billions of years of evolution; coupled with their unique mechanisms for genome plasticity, they represent an excellent system for comparative functional genomics. One example is the diversity of peptide secretion systems that modulate the interactions of bacteria with their environments. Gram-positive bacteria such as Bacillis subtilis use the general secretory (sec) pathway to translocate unfolded proteins (Ling Lin et al., 2007), and the twin arginine translocation (tat) pathway to translocate folded proteins. Gram-negative bacteria, such as E.coli, employ additional secretion systems to move proteins across their outer membranes. For example, type II secretion relies on the sec or tat pathways for translocation of proteins across the inner membrane to the periplasmic space, where they may be further modified before translocation across the outer membrane (Sandkvist, 2001). These secretion strategies may be extended by Gram-negative bacteria to engage host cells. Type III secretion systems, which are not sec-dependent, allow bacteria to inject effectors, including toxins, directly into eukaryotic host cells rather than secreting them into the extracellular milieu (Coburn, Sekirov & Finlay, 2007). Type VI secretion systems allow direct delivery of target proteins to prokaryotic or eukaryotic cells, but they are only sometimes associated with pathogenicity and are structurally quite distinct (Kulp & Kuehn, 2010; Silverman et al., 2012; Tseng, Tyler & Setubal, 2009). In this case study, we assess functional differences in these diverse peptide secretion systems based on a cross-species pathway comparison in B. subtilis, which is Gram-positive, and two Gram-negative bacteria: a non-pathogenic strain of E. coli, and P. aeruginosa.

Gene expression and gene interaction data were readily available for all three of these bacteria. ARepA provided an easy and reproducible way to integrate these data types (Tables 2A and 2B). These large datasets were integrated into three species-specific functional networks using a shared gene identifier (UniRef90), which we used to investigate genes highly associated with secretion by extracting high-confidence subgraphs around the sec and tat genes and the type II, type III, and type VI secretion systems (see Methods, Table S4). Because Gram-positive bacteria such as B. subtilis use only the general secretory (sec) and tat pathways, we extracted a high-confidence subgraph from the B. subtilis-specific functional network around two tat genes and nine sec genes, resulting in a high-confidence subnetwork containing 16 genes in total (Fig. 4A). In contrast, the Gram-negative E. coli employs an additional secretion system (type II secretion system), and we extracted a high-confidence subgraph from the E. coli-specific functional network around four tat genes, twelve sec genes, one type I gene, and twelve type II genes (see Table S4), resulting in a high-confidence subnetwork containing 34 genes in total (Fig. 4B). P. aeruginosa, as a Gram-negative with pathogenic potential, additionally utilizes type I, III, V, and type VI secretion systems, and we extracted a high-confidence subgraph from the P. aeruginosa-specific functional network around three tat genes, nine sec genes, two type I genes, eleven type II genes, twelve type III genes, one type V genes, and six type VI genes (see Table S4), resulting in a high-confidence subnetwork containing 49 genes in total (Fig. 4C).

These resulting integrated molecular networks easily recover the highly functionally related gene clusters within each individual system, in addition to interconnecting the systems themselves through a combination of conservation and co-function (Fig. 4). As the sec and tat pathways are highly conserved, we observed their corresponding clusters in all these bacteria to be highly functionally connected. The type II secretion pathway, which depends on the sec pathway in Gram-negative bacteria, is functionally connected to it both the E. coli and P. aeruginosa networks. Likewise, the sec-dependent type V pathway (Bleves et al., 2010) is also tightly connected to the sec/tat cluster in the Pseudomonas network. In contrast, the Pseudomonas type VI secretion pathway, which is tat and sec-independent (Filloux, Hachani & Bleves, 2008), was much less tightly connected to the tat/sec cluster than was type II secretion. Type I secretion, which is sec-independent (Bleves et al., 2010), was as expected only weakly connected to tat/sec. The type III secretion pathway, which is also sec/tat independent (Kostakioti et al., 2005), is also unconnected to sec/tat in the Pseudomonas network, but it is tightly connected to genes from the type II secretion pathway instead. Our underlying data for integrating the Pseudomonas network show high, confident co-expression (normalized z-score >2.0) between some genes from these clusters (e.g., yscT (type III) and gspC (type II) (Kang et al., 2008; Rao et al., 2008), yscT (type III) and gspD (type II) (Williamson et al., 2012) (Fig. 4)), suggesting a functional relationship between the type II and type III secretion systems in P. aeruginosa.

Standardized and reproducible high-throughput data retrieval and integration

As evidenced by these studies, ARepA provides an analysis platform that is modular and hierarchical. The system includes access to multiple repositories; within each, hierarchical data retrieval first queries all experiments from a particular source, then all datasets within each experiment, and finally all metadata for each dataset, in a manner similar to the design of the ISA standard (Rocca-Serra et al., 2010). Data from each repository can be retrieved independently if desired, as can data from individual datasets, and a user can customize multi-layered data processing pipelines in a consistent, templated manner if desired. Each repository retrieves multiple experiments (typically organized by taxon ID and publication ID), and each experiment can retrieve multiple platforms or datasets. Information is shared vertically, and data passed from parent to child analyses can include taxonomic information, desired dataset types, and repository-specific configuration information, such as type of normalization functions and machine learning algorithms. An ARepA user can generally request that all modules beneath a particular repository be executed, or that only a single submodule (e.g., one dataset) is processed.

As an example of dedicated module implementation, the GEO module first downloads raw gene expression data (SOFT or SeriesMatrix files) as well as platform annotations from the Gene Expression Omnibus (Fig. 5). The number of platforms used by each dataset is determined, and data for each platform is built recursively. The expression data is parsed, normalized and imputed (Huttenhower et al., 2008), and gene IDs are standardized using GeneMapper. Finally, metadata is created for each dataset, combining automatically-derived information from the SOFT files with manually curated tables if available. The final outputs are a standardized and consistent gene expression matrix encoded as tab-delimited text, corresponding metadata, a normalized co-expression network, and an R data file containing an expression set. Other dedicated modules, such as those for interaction data, operate similarly by first downloading and parsing the requested data and directly generating tab-delimited text networks for each dataset (typically separated by taxon ID and publication ID). Standardization of interaction data and metadata are performed as described for the GEO module.

Figure 5 Analysis and processing steps available for datasets from each data source.

The main steps of ARepA are divided up into four components: (1) Configuration and data integration: optional user-provided information can be merged with default data/metadata from the repositories. This allows, for example, integration of expert curated metadata with automatically annotated metadata. (2) Custom data processing, including the default and customizable gene mapping and metadata annotation, as well as processes for file format detection and conversion. (3) Data normalization: gene identifiers are standardized, gene expression levels are normalized (e.g., log-transformed), missing values are imputed using k-nearest neighborhoods, and duplicate entries are merged. (4) Data export: data file formats are normalized to tab-delimited text, and co-expression networks in text and binary formats are constructed. Gene expression datasets and automatically generated documentation are further compiled into an R data file.

Finally, data retrieved in this environment can be integrated or meta-analyzed either heterogeneously (e.g., our bacterial study) or homogeneously (e.g., the human prostate cancer network integration and murine gut differential expression studies) by procedures such as fixed or random effects meta-analysis or network integration. The former can be directly applied to identify which genes are consistently up- or down-regulated in a given organism under selected experimental conditions. This environment facilitates the identification of relevant experiments, collection of associated datasets, and application of standardized R/Bioconductor meta-analysis environments such as metaphor (Viechtbauer, 2010) to gene expression data from various laboratories and platforms. Network integration studies, in contrast, can be performed for data from any source of interest (e.g., co-expression, protein–protein interactions, regulatory interactions, posttranslational modifications, etc.) by, effectively, meta-analyzing gene interactions (edges) rather genes (nodes). The simplest approach averages across normalized interaction scores (as illustrated in the human prostate cancer network integration and bacterial study) and can be extended using more sophisticated probabilistic (Franceschini et al., 2013; Lee et al., 2004; Myers & Troyanskaya, 2007; Zuberi et al., 2013) or discriminative (Chikina et al., 2009; Guan et al., 2012) models. Both gene and interaction level integrations can be performed from within ARepA directly from the standardized metadata and formatted data.

ARepA data model and availability

ARepA includes default outputs for a variety of model organisms (Tables 1 and 2) in several standardized formats, including tab-delimited expression matrices, gene interaction networks, and R data files (see Methods). For three eukaryotic models (human, mouse, and yeast), we have chosen UniRef100 as a standardized gene identifier; this UniRef model combines identical sequences into a single family, while for bacteria the system provides UniRef90 which are built by clustering UniRef100 sequences at 90% sequence identity (Suzek et al., 2007). As expected, the largest amount of gene expression data is currently retrieved for human (>10 K datasets comprising ∼500 K samples) and for mouse (∼4 K datasets comprising >50 K samples), while among the prokaryotes, most gene expression data was retrieved for E. coli (184 datasets comprising >2 K samples). In terms of gene interaction data, ARepA by default produces the most interaction data for human (>320 K interactions), yeast (>150 K interactions), and E. coli (>100 K interactions). Functional associations across all six organisms tended to comprise more interactions (>380 K) than did physical protein–protein interaction data alone (>250 K). In terms of physical interactions, BioGRID yielded twice as many physical interactions as IntAct across the eukaryotes, while most of the physical interactions in the three prokaryotes were retrieved from IntAct. Since Bacteriome and RegulonDB are databases for physical and regulatory interactions specific to E. coli, their contribution to physical and regulatory interactions across species was limited, and the physical interactions derived from Bacteriome make up more than 50% of all physical interactions in E. coli.

Discussion

ARepA was developed to provide an automatic, consistent, reproducible, and high-throughput method of accessing, retrieving, standardizing, processing, and integrating diverse, heterogeneous data resources, suitable for large integrative analyses and meta-analyses in translational and bioinformatic research studies. To illustrate ARepA’s utility in such studies, we have described here three biological studies demonstrating the utility of ARepA for integrative analyses. When focusing on human data, ARepA’s metadata database allowed us to identify and standardize ten human prostate cancer gene expression datasets from GEO, which were subsequently meta-analyzed, while a subsequent co-expression network analysis recovered new candidate genes with potential roles in the NFκB signaling pathway in prostate cancer. A similar example in mouse integrated six gene expression datasets selected by querying ARepA for metadata indicating germ-free and intestinal tissue conditions. Finally, multiple data types from three model microbes were integrated to assess functional differences in peptide secretion systems.

In this study, we correctly recovered and extended the molecular network of the NFκB signaling pathway in human prostate cancer by using this environment to identify ten tissue-specific human expression datasets and subsequently meta-analyze them across six different array platforms. The total computing time needed to identify, normalize, and integrate these datasets was approximately 1 h. In contrast, performing a comparable process using Taverna (Hull et al., 2006) required 8 total main steps: (1) downloading gene expression data from GEO, (2) loading and accessing gene expression data (maxd software: maxdLoad2 and maxdBrowse (Hancock et al., 2005)), (3) attaching (curated) metadata (maxdLoad2), (4) normalizing and pre-processing the raw data (Taverna/Biomoby/runPreprocessing and AffyArrayNormalization (Kawas, Senger & Wilkinson, 2006) or RShell (Wassink et al., 2009)), (5) identifying and downloading dataset specific mapping files, (6) mapping from probeset to gene identifier (Taverna/Biomoby/GeneExpressionAnnotation, BeanShell scripting language), (7) generating co-expression networks (Huttenhower et al., 2008), and (8) building an integrative expression set object. Running a similar process using GEOquery (Davis & Meltzer, 2007) required seven total main steps: (1) downloading expression data from GEO (R/GEOquery/getGEO), (2) normalizing and pre-processing the raw data (R/affy/RMA (Gautier et al., 2004; Irizarry et al., 2003)), (3) identifying and downloading dataset specific mapping files, (4) mapping from probeset to gene identifier for non-standard microarray platforms, (5) attaching (curated) metadata, (6) generating co-expression networks (Huttenhower et al., 2008), and (7) building an integrated expression set object. Using ARepA requires none of these steps, as it is specifically constructed for acquisition, standardization, query, and integration tasks, resulting in co-expression/interaction networks and expression sets without the need for additional intervention.

Performing this task in ARepA allowed us to recover several genes already associated with NFκB and prostate cancer (Davis, Kucuk & Sarkar, 1999; Perkins, 2012), in addition to original findings including MEN1 as a novel putative upstream regulator (Bouwmeester et al., 2004; Hacker & Karin, 2006; Heppner et al., 2001; Wen et al., 2000) and ACBD6 as a potential downstream suppressor of NFκB-activation (Oberti et al., 2011; Pringle et al., 2012; Soupene et al., 2012; Wu et al., 2006). MEN1 was linked to cancer, prostate cancer and NFκB by two main data sources. First, MEN1 co-immunoprecipitated with the NFκB proteins p50, p52, and p65 in vitro and in vivo, revealing MEN1 as a repressor of NFκB/p65-mediated transcriptional activation and a tumor suppressor protein (Heppner et al., 2001). Second, MEN1 and NFκB/p65 were co-expressed in hepatocellular carcinoma (HCC) tissues, adjacent normal liver tissues and various cell lines (including THLE-3, HepG2, Hep3B, HuH-7 and Li-7 cells) (Gang et al., 2013). Although no mechanism was previously proposed, heterozygous Men1 mutant mice (Men1±) carrying an inactivated Men1 allele (Bertolino et al., 2003) and mixed with C57BL6/129-Sv mouse strains developed prostate cancer as compared with wild-type (Men1+/+) littermates in aged mice (18 to 26 months) (Seigne et al., 2010). The strong functional linkage of MEN1 with IKBkG in our data, as well as p53, supports the previously suggested role for this gene product upstream of the NFκB pathway as a tumor suppressor by inhibition of NFκB-mediated transactivation. This could be further validated by, for example, siRNA knockdown of MEN1 in LAPC4 cells to determine whether MEN1 silencing results in more rapid cellular growth in a hormone-depleted state.

To the best of our knowledge, ACBD6 has not been directly linked to cancer, although its expression promotes C. trachomatis-mediated carcinogenesis in vitro (Soupene et al., 2012). A homologous protein also carrying the acyl-CoA binding domain, ACBD3, was highly expressed in primary pigmented nodular adrenocortical disease (PPNAD) tissues (Liu, Li & Papadopoulos, 2003), lung cancer tissue in treated patients that did not respond to gefitinib (Kakiuchi et al., 2004), colonic mucosa cancer tissue upon pretreatment versus post-treatment with celecoxib (Glebov et al., 2006), and bone tissue in osteosarcoma patients (Jones et al., 2012), supporting a role as a potential oncogene (Fan et al., 2010). Additionally, another homologous protein, ACBD4, was highly up-regulated in cell lines from different tumor types (including MCF-7 breast adenocarcinoma, HeLa cervical cancer, HT1080 sarcoma, KB laryngeal carcinoma, SW480 colon carcinoma, Hep-2 oral carcinoma and D54 glioma) after inducing by the histone deacetylase inhibitor valproic acid (VPA) (Chavez-Blanco et al., 2006), while Acyl-CoA synthetase (ACS), which converts long-chain fatty acids into acyl-CoA, is overexpressed in different cancer cell lines (including glioma SF268 and U251 cells) (Mashima et al., 2009), lung cancer (Pei et al., 2013), and glioblastoma (Pei et al., 2009). Based on such links between acyl-CoA binding domains, their synthetase, and cancer, as well as our data that predicted a strong functional link between ACBD6 and NFκB via its inhibitors NFκB IA and NFκBIE (Pringle et al., 2012; Wu et al., 2006) (Fig. 2), we suggest a potential role of ACBD6 in prostate cancer as an upstream suppressor protein of NFκB activation. This could be experimentally validated by screening series of prostate cancer cells (LAPC4, LNCaP) to determine whether ACBD6 is present, and in those cell lines in which it is present, siRNA knockdown of ACBD6 could be performed do to functional assays, such as proliferation in hormone sensitive (LNCaP) and hormone resistant cell lines (LNCaP-abl).

Our study of host-microbiota interactions focused on identifying transcripts differentially expressed in the germ-free mouse colon. As expected based on prior knowledge of germ-free mice, there was strong differential expression of immune and carbohydrate metabolism-related genes. Meta-analysis showed that genes from the Ppar signaling pathway were highly differentially expressed under germ-free conditions. Specifically, Ppar-α, a major regulator of lipid metabolism was upregulated. Ppar signaling overall plays key roles in regulating energy metabolism, cellular proliferation and differentiation in many tissues, including the intestine, kidney, heart, and lung (Braissant et al., 1996; Nephew, Polek & Khan, 1996). Dysregulated Ppar signaling has been implicated in metabolic syndrome, for which there is an increasing appreciation for the role of gut microbiota (Aitken & Gewirtz, 2013; Tremaroli & Backhed, 2012). Prior work using Ppar-α deficient mice coupled with gnotobiotic re-derivation suggests that Ppar-α does not function in the microbiota-dependent regulation of fat storage (Backhed et al., 2004). Thus to determine its specific role in this predicted context, follow-up work could include knockout mice for these genes (inclusive of conditional mice-deficiencies within intestinal epithelial or immune cells within the gut), with or without defined microbial consortia, screened using different diets (e.g. high fat, methionine and choline deficient). The observation that the germ-free state resulted in significant perturbations to Ppar signaling pathways suggests that how microbial consortia and their metabolites affect host regulation of obesity-related inflammation, insulin responsiveness and cholesterol homeostasis should be further investigated.

In the third study, multiple data types from three model microbes (B. subtilis, E. coli, and P. aeruginosa) were integrated to correctly identify differentially conserved peptide secretion system pathways in the Gram-positive bacterium B. subtilis and the two Gram-negative bacteria E. coli and P. aeruginosa. The sec and tat pathways are highly conserved across all three bacteria and were easily retrieved during this analysis, as were type II secretion in E. coli and P. aeruginosa and the type I, III, and V, and VI pathways in P. aeruginosa (Kostakioti et al., 2005). The E. coli network was built from expression data from strain K12 (taxonomy ID 83333). While some E. coli strains (e.g., strain O157:H7) have type III secretion systems (Sharma et al., 2012), strain K12 does not. Although it has a (type I) TolC gene, it also lacks a complete HlyABCD type I secretion operon (Bakkes et al., 2010). We populated our K12 expression-based network with only the secretion systems possessed by strain K12, and correctly recovered strong sec/tat co-expression, and strong sec/tat/type II co-expression.

In P. aeruginosa, there was a strong type II/III secretion link as well as strong tat/sec coordination, while type VI secretion was only weakly connected to sec/tat. Type V secretion, which is sec-dependent (Bleves et al., 2010), was strongly linked to sec/tat. The strong coregulation of type II and III secretion in Pseudomonas is potentially due to their joint role in pathogenesis; for example, in Pseudomonas-mediated pneumonia, type II secretion is used to secrete proteases such as LasA and LasB, phospholipase H, and lipolytic enzymes which degrade host tissue; type II secretion allows Pseudomonas infections to persist in mice (Jyot et al., 2011). The toxins secreted by the type III secretion system are complimentary, but focused on immune inhibition rather than tissue destruction. For example, ExoT slows wound healing and inhibits bacterial internalization by macrophages, and ExoS induces lymphocyte apoptosis (Schulert et al., 2003). Pseudomonas that lacks type III secretion is much more vulnerable to immediate clearance by neutrophils (Sun et al., 2012). Thus, the type II and type III secretion systems in Pseudomonas work together to help it persist in the host in infection. Type VI secretion in Pseudomonas is much less well-characterized, but proteins secreted by this mechanism do not contain sec or tat signals (Pallen, Chaudhuri & Henderson, 2003). While type VI secretion has been implicated in biofilm production (Zhang et al., 2011), the operon associated with biofilm is repressed by quorum sensing (Bleves et al., 2010), so an association with sec and tat that is driven by quorum sensing for the purpose of biofilm production is unlikely.

In comparison with existing bioinformatics workflow tools like Taverna (Hull et al., 2006) and Galaxy (Giardine et al., 2005; Goecks, Nekrutenko & Taylor, 2010), ARepA is specifically tuned to download, normalize, and meta-analyze very large and diverse collections of high-throughput biological data. While users can author new modules for other analyses of the resulting integrated data, the system is tuned to facilitate this task seamlessly out of the box. Galaxy, on the other hand, focuses primarily on real-time, interactive analyses, typically of a few datasets of interest through a graphical user interface. This makes it very flexible when developing new or customized analysis workflows of particular datasets, but less easily adaptable to bulk processing of especially diverse data. Taverna, alternatively, provides an extensive platform for combining bioinformatics tools in a graphical environment, but again it does not directly facilitate bulk data download, normalization, and integration. By uniformly normalizing and providing metadata for results deposited across many high-throughput data repositories, ARepA makes it easy to integrate all relevant data types for organisms, phenotypes, diseases, or molecular targets of interest.

One of the most important application areas for an environment such as ARepA is in translational meta-analyses, which relies on large data collections from multiple sources to ensure cross-study validation (Ganzfried et al., 2013; Griffith et al., 2006). ARepA supports such efforts in two ways. First, it effectively integrates data of heterogeneous format and origin without the need for manual curation or user intervention. This is true both for data normalization and for metadata collection, the latter of which is also integrated into a single, searchable database from across multiple repositories. ARepA’s automated processing can thus be used as a first step in particularly highly targeted manual curation if desired, as has been the case in our own previous studies (Ganzfried et al., 2013; Waldron, Coller & Huttenhower, 2012). Second, ARepA provides complete provenance and reproducibility for all data retrieved, as this is similarly a priority in pre-clinical research. The data processing and normalization steps carried out by ARepA can be modified or added to by a user if desired, ensuring that exactly the same steps are carried out at each execution for each dataset. This remains practical even when handling large data collections since ARepA’s dependency-tracking architecture ensures execution only when the input data source has changed or new analyses are added.

A second potentially important environment for ARepA is as a basis for “in-house” repositories for groups supporting internal data or focusing on specific types (e.g., stem cell data). This provides a convenient way to start from public repositories by first screening metadata for areas of interest. This screen can be added to ARepA’s data retrieval process for each source repository, ensuring that only data of interest is downloaded and updated each time; the latter is important since public repositories do updated regularly and on a varying basis (e.g., GEO is updated continuously, BioGRID once a month, and STRING once to twice per year). Privately maintained data can be appended by adding a new module to ARepA that fetches data locally, allowing it to be standardized to the same format as that of the other repositories for integration. ARepA is generally designed to be customizable to individual user environments, allowing modification of data retrieval or processing scripts while still taking advantage of its dependency-tracking system. Public data are typically fetched via ftp or http; however, a user can easily acquire data using other protocols such as ssh, Aspera, or simple local copies or links. This flexibility in design allows users to integrate publically available data with privately maintained data to conduct in-silico experiments within a single consistent environment.

Since any high-throughput genomic assays are susceptible to batch effects (Leek et al., 2010), analysts performing meta-analyses should be aware of their potential impact. Batch correction methods in general either require a priori knowledge of the batches (e.g., for empirical Bayes adjustment (Johnson, Li & Rabinovic, 2007)) or are specific to the outcome variable of a particular differential expression (Leek et al., 2012), as has been previously reviewed (Chen et al., 2011; Lazar et al., 2013). Thus, batch correction is a study-specific issue that cannot be resolved solely by a high-throughput tool such as ARepA, although it provides a framework within which individual datasets can be easily handled. Although study-specific sensitivity analysis of batch correction is warranted when possible, several studies have found that traditional meta-analysis approaches are robust to batch effects and that explicit batch correction may be unhelpful for independent validation of differential expression (Re et al., 2014) or prediction models (Riester et al., 2014; Waldron et al., 2014).

Like any computational environment, this is one of a number of routes available for future expansions to ARepA. An obvious next step is the inclusion of additional repository modules, with large and well-annotated sources such as ArrayExpress (Rustici et al., 2013), Refseq (Pruitt, Tatusova & Maglott, 2005), ENCODE (Consortium, 2012), or RNA-sequencing data representing excellent candidates. This will likewise allow the integration of additional data types such as genomic sequences, regulatory elements, and high quality expression data that would broaden the set of data to be considered for subsequent meta-analyses. The addition of a graphical user interface will make ARepA more accessible for a broader user community and increase its usability, as will a unified configuration interface (e.g., a single, XML-based configuration file) to more easily change parameters. The current version of ARepA thus represents only the first step in the ongoing development of an environment for reproducible “big data” research, which we hope will be both expandable by bioinformaticians and useful for targeted biological studies.

Supplemental Information

Figure S1 Number of murine gene expression datasets from GEO grouped by experimental types, disease models, and tissue types

We screened ARepA’s murine metadata by matching different experimental types, disease models, or tissue types and obtained different lists of matching datasets for which the number of datasets are visualized in this figure.

Click here for additional data file.

Table S1 A set of ten human prostate cancer specific gene expression datasets from GEO

We screened ARepA’s GEO metadata database for human prostate cancer and prostate tissue conditions matching “prostate cancer” in human gene expression profile studies with at least six samples each, identifying ten relevant datasets from six different platforms (Affymetrix, Agilent, and CNIO).

Click here for additional data file.

Table S2 A set of six murine case-control gene expression datasets from the intestinal tissue from GEO

We identified six case-control datasets containing intestinal tissue and germ-free versus wild type comparisons by screening ARepA’s murine metadata by matching “germ-free”, “wild type”, and “intestinal tract” in mouse gene expression profiles with at least four samples each.

Click here for additional data file.

Table S3 Enriched pathways from a meta-analysis of murine gene expression datasets from germ-free vs wild type intestinal tissue

We conducted a differential expression meta-analysis of genes and pathways up- or down-regulated in the germ-free murine gut (Figure S1), using six case-control datasets (Table S2). We first computed differential expression for all genes between germ-free gut and wild type gut within each dataset using the R/limma package. Next, we performed a meta-analysis using the R/metafor package on these six datasets by applying a random-effects model on the fold changes, fitting the model with the restricted maximum-likelihood estimator (REML). We finally tested all genes and their resulting meta-p-values for gene set enrichment in KEGG and BioCarta pathways which are shown in this table.

Click here for additional data file.

Table S4 Lists of query genes for extracting high-confidence subgraphs in human and bacteria

To extract high-confidence subgraphs from global networks, we defined lists of query genes for each context. First, we defined a list of 12 query genes from the NFκB signalling pathway in BioCarta for extracting a high-confidence subgraph from the prostate cancer specific co-expression network in human. Second, we defined three lists of query gene sets of the sec and tat genes and genes from the Type II, Type III, and Type VI secretion systems for one Gram-positive (B. subtilis) and two Gram-positive bacteria (E. coli and P. aeruginosa).

Click here for additional data file.

Table S5 Meta-analysis of murine gene expression datasets from germ-free vs wild type intestinal tissue

We conducted a differential expression meta-analysis of genes and pathways up- or down-regulated in the germ-free murine gut (Figure S1), using six case-control datasets (Table S2). We first computed differential expression for all genes between germ-free gut and wild type gut within each dataset using the R/limma package. Next, we performed a meta-analysis using the R/metafor package on these six datasets by applying a random-effects model on the fold changes, fitting the model with the restricted maximum-likelihood estimator (REML).

Click here for additional data file.

Text S1 Details of ARepA case studies

Click here for additional data file.

Text S2 Metadata file example

Click here for additional data file.

Supplemental Information 9 Scripts for reproducing the case studies

Click here for additional data file.

Additional Information and Declarations

Competing Interests

Author Contributions

The authors declare there are no competing interests.

Daniela Börnigen conceived and designed the experiments, performed the experiments, analyzed the data, wrote the paper, prepared figures and/or tables, reviewed drafts of the paper, implemented software.

Yo Sup Moon performed the experiments, wrote the paper, prepared figures and/or tables, implemented software.

Gholamali Rahnavard and Lauren McIver implemented software.

Levi Waldron conceived and designed the experiments, performed the experiments, analyzed the data, wrote the paper, reviewed drafts of the paper, implemented software.

Afrah Shafquat contributed reagents/materials/analysis tools, wrote the paper.

Eric A. Franzosa contributed reagents/materials/analysis tools, prepared figures and/or tables, reviewed drafts of the paper.

Larissa Miropolsky conceived and designed the experiments, implemented software.

Christopher Sweeney and Wendy S. Garrett analyzed the data, reviewed drafts of the paper.

Xochitl C. Morgan analyzed the data, wrote the paper, reviewed drafts of the paper.

Curtis Huttenhower conceived and designed the experiments, analyzed the data, wrote the paper, reviewed drafts of the paper.

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
