# Peer review of "A reproducible approach to high-throughput biological data acquisition and integration"

_PeerJ, doi:10.7717/peerj.791_

## Round 0.1 · original submission · Major Revisions

Three highly qualified reviewers have now assessed your submission and as you can see below, all three are generally enthusiastic about it. I share their enthusiasm and think that this work could be of high interest to the community.

Inevitably, however, some important points will need to be addressed prior to publication. In particular, given the (timely and highly relevant!) focus on reproducibility, the work should strive to meet the highest standards in this respect. Reviewer 1 and 3 provide detailed feedback that should help you achieve this. I urge you to address these in earnest.

I think that the illustrative case studies are valuable and sufficiently compact, so I do not see the necessity to shorten them (as suggested by reviewer #1). The figures don't look truncated to me so I suspect this may be a problem with the referee's PDF viewer.

Reviewer 1 ·

Basic reporting

The structure and text of the manuscript are acceptable; I have no significant comments. However, perhaps too much space is dedicated to discussing the biological findings of the case studies; this detracted from the communication of the core message of the manuscript.

This may have been an artifact of the upload process, but figures 1, 3, 4 and 5 appear to have been significantly truncated.

Experimental design

My comments mostly concern the implementation of the software.

It should be noted immediately that the software did not run on my PC (GNU/Linux, Python 2.7.7, all other software requirements met). This was true both for the source archive available from the project's webpage and for the most recent revision in the source repository:

Traceback (most recent call last):
File "/opt/arepa/src/taxdump2taxa.py", line 43, in <module>
if iLevel <= iHit:
TypeError: unorderable types: int() <= NoneType()

Thus, I could not test that the system meets the authors' claims.

It is never made clear why the software outputs metadata in a Python-specific serialized (pickle) format. If this is a significant feature of the system, it should be more properly explained; otherwise, a plain text file (perhaps JSON or XML format) would be much more flexible.

Gene expression data normalization is performed via Seipnir. This method should be given a brief overview in the text, otherwise it is easy to miss that reference and to be left wondering how the normalization that is referred to throughout the text was done. This is important for being able to correctly interpret the results.

I would also have found it useful to have a brief explanation of the role of SCons in the system already in the Implementation section. I had to surmise its purpose by reading the rest of the manuscript.

Looking at the code, I find very little exception handling (only fourteen "try/except" blocks in the whole system) or other signs of careful design to prevent or mitigate user error. Thus, I am immediately skeptical of any claims of robustness of the system (e.g. line 143 in "Introduction"). Given that the system is designed to allow user configuration and extension, small mistakes could lead to unhandled exceptions or other unexpected behavior that might generate erroneous data. If the aim is for ARepA to be robust in its reproducibility, the authors must guarantee so in their code.

Nowhere is it mentioned which versions of the various databases were used to perform the case study analyses. Considering the theme of reproducibility, this is a lamentable oversight.

I have no additional comments on the experimental design of the case studies, which appear to be otherwise biologically sound.

Validity of the findings

While I agree with the central motivation of the manuscript, that reproducible methods for retrieving and integrating data from heterogeneous sources are essential, I do not think that ARepA satisfies this in its current implementation. The need for such reproducibility comes in two forms: for the individual researcher/group, to ensure that subsequent runs of the same analysis use the same data, and for external researchers/groups who seek to reproduce the findings of others. When it is run by an individual researcher on the same computer, it can reasonably be expected to give reproducible output. However, the system in no way ensures or even verifies that an external researcher will have the same results. Likewise, I see no indication that reproducibility of an analysis will be maintained over long periods of time, as software and databases are updated.

In its current form, the system does, indeed, provide a modular, extensible design that should enable convenient retrieval and integration of data from diverse sources. With regards to reproducibility, though, I'm afraid that it doesn't provide any stronger guarantee than a simple shell script with data source URLs hard-coded in it would. On the other hand, if one does want to provide reproducibility in a flexible manner, an architecture like the one provided by the current ARepA design is a necessary foundation. Thus a couple critical features come to mind that must be implemented before the system can begin to be considered to provide reproducibility.

The most essential feature is that the system must provide a method of actually verifying that the data output by two different runs of the software are the same. How can reproducibility be guaranteed by software when data consistency cannot even be easily checked within it? An obvious method would be to provide a hash/checksum, which then can be easily compared between runs. This can be done manually by the user, but it really should be a core feature of the software. The hash should include both the input and the output; that is, the same set of parameters should always generate the same output, and this must be reflected in a consistent hash (alternatively, separate hashes of input and output could be useful).

The second, which is related, is actually mentioned in the final paragraph of the manuscript, what the authors refer to as "version control" of the analysis software/scripts (although I would prefer a different term, since this usually refers to source control mechanisms like git or SVN), which they mention as a potential future development. Actually, this is an absolutely essential feature, especially if ARepA is to see continued core development by the authors or the addition of extensions or modifications by users. The software should also similarly track the versions of the databases that it accesses (these versions should really be first-class parameters, directly editable in the configuration files rather than being hard-coded in the URLs). Without a method to track these differences, one cannot begin to assess the potential sources of inconsistencies in the final output.

With respect to the case studies, the results do a good job of illustrating the potential of the system, especially in its flexibility. That the authors even find new interactions in the human prostate cancer case is a testament to the potential of the tool for meta-analysis. However, the case studies do nothing to test or otherwise demonstrate the reproducibility of the output, which I understand to be the core argument being made for the software. Indeed, it would be difficult to make such tests without the missing features mentioned above.

Additional comments

In general, I think the concept behind ARepA is sound and software like it is sorely needed in science, but it does not yet achieve robustness and reproducibility promised in the manuscript. More attention should be given in the manuscript to testing these features of the software in addition to demonstrating potential analyses using it.

A key question is whether someone reading about a study in which ARepA was used, two years after that study was published, would be able to run the software and get the same results. Of course, directly testing and demonstrating this would be the case is not feasible, however a number of issues could be addressed (as described above) that would lend confidence to the assertion of consistent and reproducible output.

I have a couple of additional nitpicks:

- Throughout the manuscript, the authors refer to ARepA producing an "R package" (e.g. line 229 or 364). Typically "R package" would refer to a library that includes functions, whereas I am under the impression that in this case, it only includes data that is intended to be attached in an R session. Thus, it might be more accurate to refer to it as an "R data file" (especially since "R package" is used in its usual sense on line 769).

- It is not my intention to comment on the code outside of how it affects the science reported in the manuscript. However since it is expected to be modified and extended by the user, I feel obliged to point out that, unlike most languages, Python has a strong set of coding style guidelines that go along with it (http://legacy.python.org/dev/peps/pep-0008/), which ARepA does not follow. While it is not strictly necessary to adhere to them, to do so would help others who seek to modify the system for their own purposes. That said, it is a verifiable problem in at least one file (Bacteriome/src/bacteriome.py), in which tabs and spaces are mixed for indentation, which is illegal on newer Python versions, including 2.7.7.

Reviewer 2 ·

Basic reporting

The references do not appear to be correctly formatted.

Experimental design

No comment.

Validity of the findings

No comment.

Additional comments

1. “Each of these interfaces is, unfortunately, database-specific; expression data retrieved programmatically in R must be manually converted and combined with interaction data downloaded through a web interface” – this is somewhat misleading. IntAct, BioGRID, STRING and MPIDB all make their data available, and downloadable in the PSI-MI standard formats. There is also an existing R interface to that format – see PMID: 19325550 for further details.

2. “Ensembl BioMarts (Kinsella, Kahari et al. 2011) , are among the
most common. Each is customized to automate queries and downloads of one particular database,” – in fact queries can be written across two BioMarts i.e. across 2 databases.

3. “These were processed as above and standardized onto a shared gene identifier (UniRef90).” – whilst I accept UniRef90 will largely cluster at the gene level, the cluster is based on the protein sequence and so duplicated genes, for example, will be found in the same cluster. Identical genes will still be merged in UniRef100. This should be made clear.

4. The authors comment on the update cycle of the resources from which they draw information but I could not see how often the public implementation of ARepAis updated. The numbers for IntAct given in the online documentation appear to be particularly out of date – since the merger with MINT (PMID:24234451) this database contains nearly 500,000 physical interactions. The section analysing the data content of each repository should indicate when the download for this analysis was made.

·

Basic reporting

No Comments

Experimental design

No Comments

Validity of the findings

No Comments

Additional comments

Borningen et al, in their manuscript "A reproducible approach to high-throughput biological data acquisition and integration" describe the ARepA framework to automate the query, collection, download and processing of various omics data. They illustrate the tool with three diverse and well described case studies: expression data from pancreatic expression data, host-microbiome interaction in germ-free mice and secretion pathways in E. coli, B. subtilis and P. aeruginosa.

I think that this work is an elegant and a useful addition to the tool set of available software for integrative data analysis. The manuscript is clearly written and laid out. The software is freely available under a permissive open source license, the source code itself is version-controlled and distributed through a public bitbucket repository and the project is well documented. Below, I provide comments and suggestions for the authors. These points are roughly presented in order of importance, starting with the more highly ranked ones.

- Reproducibility is indeed essential, as highlighted by the authors. It would be particularly useful if they gave the means to reproduce the case studies presented in the paper by providing, for example, the necessary configuration files or a complete ARepA tar ball after execution. This would in turn facilitate the testing and the adoption of ARepA by others. Distributing the resulting data sets, as text files and R packages would also allow users to validate their understanding of the tools and their execution pipeline for the given input.

- An important aspect of any pipelines, and arguably even more those that aggregate data from multiple sources, is the assessment of the quality of the data. Have the authors experimented with the inclusion of dedicated Bioconductor package such as, for example, arrayQualityMetrics for expression data and produce a standardised QC report in addition to the processed data?

- The manuscript is generally well written and clear. However, I felt that it was a times repetitive. For example, data sources, the description and summary of the use cases are reiterated several times. This could be rationalised and the implementation of ARepA could be elaborated on and possibly illustrated with a figure demonstrating the hierarchical nature of the framework. In particular, the implementation and customisation of the querying mechanism was not clear to me. In the manuscript, the authors mention the GEO search queries they have used ("prostate cancer" and "germ-free", "wild type", and "intestinal tract"), but it is not clear from the text (nor have I found this in the documentation), how to effectively implement these meta-date queries. This last point would for example be addressed through my first comment.

- The authors provide timings, but these are not informative without the specifications of the platform.

- I would appreciate a few more details about R package generated by ARepA: do they contain documentation, detailing for example the provenance and processing of the data? This would also be addressed if I could access and example package from the use cases.

- In the Material and Methods/Case studies section, the results are summarised and result figures are presented for the two first cases, while the results of the last example are presented much later in the manuscript. This could possibly be harmonised.

- Among the future expansions enumerated at the end of the manuscript, have the authors considered offering ARepA as a service, allowing users to configure and send queries on a remote server and obtain the outputs.

- While this point might be addressed in your references, I was curious about how to determine the neighbourhood size (k=5 in one case, k=10 in another)? If convenient, this could possibly be briefly addressed.

---

## Round 0.2 · accepted · Accept

Thank you for thoroughly addressing the reviewers' comments. The reviewers are satisfied and so am I. Referee #1 found a typo and has a few helpful suggestions, which I encourage you to consider. To speed things up, I am accepting the manuscript, but please address these remaining points at proof stage.

Reviewer 1 ·

Basic reporting

I find the updated structure of the manuscript to strike a much better balance between the description of ARepA tool and the biological case studies performed with it.

Experimental design

My primary concerns about the description of the methods employed by ARepA, with regards to explaining SCons, Sleipnir and the use of the pickle format have all been addressed. I find the overall process to be much easier to understand from the manuscript text now.

I understand the argument that the binary pickle format provides performance improvements over something like JSON or XML. However, I still have some reservations about using this format rather than something more general and not tied to a single language (in case a user wants to query the metadata in a different language), like an SQLite database. This is an implementation detail, though, and does not affect the validity of the science or the primary goals of the program, so I merely suggest it as a future improvement.

Validity of the findings

I am very happy that MD5 sums of the datasets have been added to the basic output of the program. I think this is a vital addition and goes a long way towards promoting reproducibility. I also thank the authors for implementing version tracking of the programs. This also does much to promote reproducibility.

I think it would be nice to provide the MD5 sums of the data fetched for the case studies somewhere in the manuscript, perhaps in the supplementary materials. As a reader who wants to test ARepA, the first thing I would do is try to replicate your studies; it would be nice to be able to quickly compare the MD5 sums that I get with those reported by you. I would like to imagine that such a thing would become a "best practice" of sorts, and so reporting these checksums yourselves would promote it. In fact, you may consider adding a script to the package (similar to unpickle.py, I suppose, but more limited in scope) that outputs a brief, formatted-text report of the key information that one would typically include in a manuscript (date downloaded, MD5 sum, URL, etc), to further promote the idea.

Additional comments

Because you now log the versions of the programs used by ARepA, you can remove the sentence appearing on line 754-756.

Also, there is a small typo on page 22, line 756: "The additional of a..." should be "The addition of a..."

I'm happy to report that the code runs fine for me. Thank you for looking into it.

·

Basic reporting

No comments

Experimental design

No comments

Validity of the findings

No comments

Additional comments

I thank the authors for addressing all the comments.